**www.cambridge.org/qrd**

# The low complexity linker of DNAJB6b is key to its anti-amyloid function

Timas Merkelis[1] (ORCID), Ulf Olsson[1] and Sara Linse[2] (ORCID)

[1]Division of Physical Chemistry, Department of Chemistry, Lund University, Lund, Sweden and [2]Biochemistry and Structural Biology, Department of Chemistry, Lund University, Lund, Sweden

## Research Article

amyloid inhibition; chaperone activity; protein aggregation; protein folding; low-complexity region

**Corresponding author:**
Timas Merkelis;
Email: timas.merkelis@fkem1.lu.se

## Abstract

Neurodegenerative disorders, such as Alzheimer's and Parkinson's diseases, are associated with the formation of amyloid fibrils. The DNAJB6b (JB6) chaperone greatly inhibits the disease-related self-assembly of amyloid peptides in an ATP-independent manner. The molecular basis of this process is, however, not understood. Here, we studied the low complexity linker between the N- and C-terminal domains of JB6 as an isolated 110 amino acid residue construct, to get a better understanding of the role of the composition of the intact protein. We investigate the structure and aggregation behaviour of the linker and its anti-amyloid activity in comparison with the full-length chaperone. We find that the linker contains ca. 45% α-helix and 20% β-sheet and is in itself an amyloid-like peptide that self-assembles into different structures, which are bigger than those formed by the intact chaperone, including fibrils. The isolated linker protects against fibril formation of Aβ42 as well as α-synuclein, but is less potent than the intact chaperone. Based on our results, we propose a possible mechanism behind JB6 and linker amyloid suppression relating to their self-assembly behaviour. In the intact protein, the domains serve to solubilize the linker such that the solution concentration of exposed linker is high enough to sustain its high potency against amyloid formation.

## Introduction

Molecular chaperones are proteins that facilitate and regulate protein folding within cells and prevent cell damage when proteins become unfolded or misfolded by stress (Kosmaoglou *et al.*, 2008). Molecular chaperones prevent metabolic and ageing-associated neurodegenerative diseases such as type II diabetes, atherosclerosis, Parkinson's, Huntington's, and Alzheimer's disease, associated with amyloid protein deposition (Arosio *et al.*, 2016; Chiti and Dobson 2017; Hartl 2017; Hageman *et al.*, 2010).

An important group of chaperones retard amyloid formation of other proteins, clients, and increases client solubility without using ATP. A prominent example is DNAJB6b (JB6) from the Hsp40 family of J-domain proteins, which is widely expressed in the brain (Durrenberger *et al.*, 2009; Hentze *et al.*, 2024). JB6 combines a high specificity in action, amyloid suppression with a high promiscuity in terms of client sequence and has been observed to prevent aggregation of, for example, poly-Q-peptides (Hageman *et al.*, 2010; Kakkar *et al.*, 2016; Månsson *et al.*, 2014b, Hobbs *et al.*, 2025), amyloid β peptide (Månsson, *et al.*, 2014a; Carlsson *et al.*, 2024), and α-synuclein (Aprile *et al.*, 2017; Deshayes *et al.*, 2019; Pálmadóttir *et al.*, 2025).

JB6 contains two globular domains, the J-domain (JD) and the C-terminal domain (CTD), connected by a low complexity linker. The JD is highly conserved over the DnaJ proteins and interacts with Hsp70 via a conserved HPD motif (Österlund *et al.*, 2023). The CTDs are more varied and inferred in client interactions (Österlund *et al.*, 2023). The linker of 110 amino acids, that is 45% of the protein chain, seems to play an important role in both the self-assembly and activity of the chaperone, as well as the interaction with Hsp70 (Ruggieri *et al.*, 2015; Österlund *et al.*, 2020; Österlund *et al.*, 2023; Hobbs *et al.*, 2025). This part of the protein consists of a limited set of amino acids, with over half of the linker residues being (glycine, G, phenylalanine, F, or serine/threonine, S/T), with two rather distinct halves, referred to as the G/F rich and S/T rich segments (Ruggieri *et al.*, 2015; Österlund *et al.*, 2020; Sarparanta *et al.*, 2020), although both halves are indeed G/F rich. A large fraction of the linker is hydrophobic, which likely governs both the self-assembly of JB6 and its interaction with amyloid proteins.

Low complexity regions are found in several other proteins where promiscuity or fast dynamics in target recognition and condensate formation are key to function. Common to such low complexity regions is the prevalence of G, S, and T, but also a strikingly high frequency of F or other aromatic residues, such as tyrosine. For example, the translocation through the nuclear pore complex protein relies on multiple FG-rich repeats, which mediate interactions with transport proteins and enable fast passage through the pore for specific cargo (Hayama *et al.*, 2018). Liquid–liquid phase separation and the co-condensation of intrinsically disordered

proteins are thought to mediate the formation of membrane-less organelles in cells (Banani *et al.*, 2017). RNA-binding proteins, e.g. TDP-43, contain low complexity, aromatic-rich segments of importance for self-assembly, condensate formation, and RNA-binding (Li *et al.*, 2018; Cermakova and Hodges, 2023).

JB6 forms oligomeric aggregates (Månsson *et al.*, 2014a,b) in a concentration-dependent manner with a critical aggregation concentration of around 100 nM (Carlsson *et al.*, 2023). The active forms in amyloid suppression are the dissociated JB6 subunits, rather than the oligomeric form (Carlsson *et al.*, 2024). Still, a mainly monomeric mutant lacking the linker region has been found to have reduced activity (Österlund *et al.*, 2020;). The self-assembly may thus be a consequence of the same molecular features that make it a potent amyloid suppressor (Linse *et al.*, 2021). DNAJB8, another J-domain chaperone that suppresses amyloid formation and forms oligomeric structures, contains a similar linker composed of G/F and S/T rich regions involved in the self-assembly and attractive interactions with amyloids (Ryder *et al.*, 2024).

While the action of JB6 has been discussed mainly in terms of its globular domains (Österlund *et al.*, 2023;), we here study the role of the 110-residue linker, i.e. DNAJB6b (76–186), devoid of the globular domains. We express the linker as a stand-alone peptide and monitor its self-assembly, structure, and amyloid suppression activity. The results allow us to discuss the thermodynamic basis (Linse *et al.*, 2021) and molecular driving forces of JB6 action in amyloid retardation and client solubility enhancement.

## Methods

### Linker expression and purification

The 110-residue linker, i.e. DNAJB6b (76–186), was expressed in *Escherichia coli* in fusion with the self-cleavable EDDIE mutant of nPro and purified via ion exchange chromatography, hydrophobic interaction chromatography, and size exclusion chromatography as described in the Supplementary Information, sections S1 and S2.

### Protein sample preparation

All chemicals were of analytical grade. The protein sample for cryo-TEM, CD spectroscopy, light scattering, and HPLC measurements was prepared from a purified monomeric stock of 25 μM in 4 M urea via dialysis to 20 mM phosphate 0.2 mM EDTA, pH 8.0, using Slide-A-Lyzer MINI Dialysis Devices with 3.5 kDa molecular weight cut-off from Thermo Fisher Scientific according to the producer's described protocol. Fibril formation and HPLC experiments were performed in PEGylated polystyrene half-area 96 Well plates (Corning 3881).

### Cryo-TEM

Cryogenic transmission electron microscopy (cryo-TEM) experiments were carried out using a JEM-2200FS transmission electron microscope (JEOL) at the National Centre for High Resolution Electron Microscopy (nCHREM), Lund University. The instrument had a field-emission electron source, a cryo pole piece in the objective lens, and an in-column energy filter (omega filter). Zero-loss images were recorded at an acceleration voltage of 200 kV on a bottom-mounted TemCam-F416 camera (TVIPS) using SerialEM under low-dose conditions. Samples were prepared using an automatic plunge freezer system (Leica EM GP) with the environmental chamber operated at 25 °C and 90% relative humidity. A 4 μL droplet of the sample was deposited on a lacey formvar carbon-coated grid (Ted Pella) and blotted with filter paper to remove excess fluid. The grid was then plunged into liquid ethane ($-183$ °C) to ensure rapid vitrification. Samples were then stored in liquid nitrogen ($-196$ °C) and transferred into the microscope prior to imaging using a cryo transfer tomography holder (Fischione Model 2550).

### CD spectroscopy

Circular dichroism (CD) spectra were recorded using a Jasco J-815 spectrometer with a Peltier thermostatic cell holder. The protein solution was studied in Hellma Analytics QS high precision quartz cells with 1 mm path length at a scan rate of 100 nm/min, with 10 points/nm slit width 1 nm, and a digital integration time of 1 s. The linker concentration in the samples used for CD spectroscopy was measured using HPLC.

### Light scattering

Light scattering experiments were performed using an ALV/DLS/SLS-5022F, CGF-8F-Goniometer system, using ALV7004 software. The laser used was 633 nm HeNe laser from Thor Labs USA.

### ThT fluorescence measurement with Aβ42 peptide or α-synuclein

All fibril formation kinetics experiments were monitored using a Fluostar plate reader (BMG LABTECH, Germany) using a 448 nm excitation filter and a 480 nm emission filter. For Aβ42, PEGylated 96-well black polystyrene half-area plates with a clear bottom (Corning 3881) were used, and no shaking was imposed beyond the reading of the plate. For α-synuclein, 96-well polystyrene half-area plates with a clear bottom (Corning 3880) were used with 200 rpm shaking between reads. Buffers were degassed and filtered using wwPTFE 0.2 μm 50 mm disc filters (Pall Corporation).

The Aβ42 peptide used in this work was Aβ(M1–42), recombinantly expressed in *E. coli*, from a synthetic gene with *E. coli*-preferred codons (Walsh *et al.*, 2009), purified via ion exchange and size exclusion chromatography as described (Linse, 2020) and stored as lyophilized aliquots at $-20$ °C. The monomeric Aβ42 was freshly purified by size exclusion chromatography just prior to starting each experiment as follows. A lyophilized aliquot of Aβ42 was dissolved in 1 mL 6 M GuHCl, 20 mM sodium phosphate, pH 8.0, before being injected onto a 10/300 Superdex 75 increase column (Cytiva) operated in 20 mM sodium phosphate, 0.2 mM EDTA, pH 8.0 at a flow rate of 0.5 mL/min. The peptide concentration was calculated by integrating the absorbance at 280 nm of the collected fraction in the chromatogram using an extinction coefficient of 1440 $M^{-1}$ $cm^{-1}$.

α-synuclein was expressed in *E. coli* from a synthetic gene with *E. coli*-preferred codons (purchased from Genscript, Piscataway, NJ, USA), purified via boiling and two steps of ion exchange chromatography as described (Pálmadóttir *et al.*, 2021). The monomeric α-synuclein was freshly purified by size exclusion chromatography just prior to starting each experiment as above. The peptide concentration was calculated by integrating the absorbance at 280 nm of the collected fraction in the chromatogram using an extinction coefficient of 5800 $M^{-1}$ $cm^{-1}$.

## HPLC

Centrifugations were performed using the Centrifuge 5424 from Eppendorf. The protein samples were centrifuged at 20000 rcf for 12 minutes prior to each HPLC measurement to pellet any aggregates. The reverse-phase HPLC was done using a BIOshell A160 Peptide C18 column from Sigma Aldrich and ACN with a gradient of MeOH up to 95% as the eluent in a Shimadzu LCMS-2020 Single Quadrupole LC/MS system with a Nexera XS HPLC/UHPLC system.

## Results

Here, we have studied the aggregation behaviour of a 110-residue peptide corresponding to the JB6 linker (DNAJB6b 76–186) by various methods on a time scale of two weeks. Frozen aliquots of 1 mL 25 µM linker peptide in 4 M urea were thawed and dialyzed to 50 mL of a 20 mM sodium phosphate buffer, 0.2 mM EDTA, pH 8.0 twice over 24 h, yielding a final urea concentration of 1.6 mM. Cryo-TEM was used to provide snapshots of aggregate development by freezing and imaging freshly prepared, one-week-old and two-week-old room temperature samples. The images show that aggregates of different morphologies are formed by the linker and that their relative abundance changes over time (Figure 1). In the fresh samples, the linker appears to mostly form dense clusters of material ranging from 200 to 1000 nm in diameter (Figure 1a). This result is consistent with the dynamic light scattering (DLS) data (Supplementary Figure S2), suggesting a particle average hydrodynamic radius of ≈ 300 nm. After 1 week the clusters seem to mostly disappear, and the material appears to be almost homogeneously spread throughout the 25 µM samples, with the aggregates appearing to take on a tangled coil-like shape (Figure 1b). These curvilinear structures we will refer to as squiggles. Some fibrillar structures can be seen but are heavily obscured. At the two-weeks mark, the sample appears highly homogenous with the squiggles spread throughout the sample (Figure 1c). Additional samples were prepared at 8 µM (Figure 1d) to see if any other aggregate forms are present and obscured by the dominant squiggles and to monitor changes in the sample over a longer time period (Figure 1e). After two weeks, fibril-like aggregates appear in the 8 µM sample and coexist with the squiggles. After a period of five weeks (Figure 1e), no additional changes were observed.

Regarding the shape and size of the aggregates (Figure 1f), the squiggles appear very thin, <5 nm, and are up to 200 nm long. The fibril-like aggregates appear to be 5–10 nm in width, with some thicker parts, and 200–1000 nm in length.

Since the cryo-TEM data indicate that the concentration of the linker aggregates increased over time, monomer quantification by absorbance in reverse-phase HPLC was used to confirm this finding. The solutions, prepared in the same way as the cryo-TEM samples, were centrifuged at 20000 rcf for 12 minutes to sediment the aggregated material, while the supernatant was collected, and its protein concentration was evaluated based on the UV absorbance at 205 nm. The results (Figure 2) showed an approximately exponential decrease in monomer concentration over time, with an initial concentration after dialysis of 21 µM, a half-time of 62 h, reaching a value of 63 nM.

To study the reversibility of aggregation, an aggregated 25 µM linker sample was diluted 100 times and incubated for two weeks at both 25 and 37 °C. No linker monomer was detected by absorbance in HPLC as above in either sample, measuring several times over the period, implying that the monomer

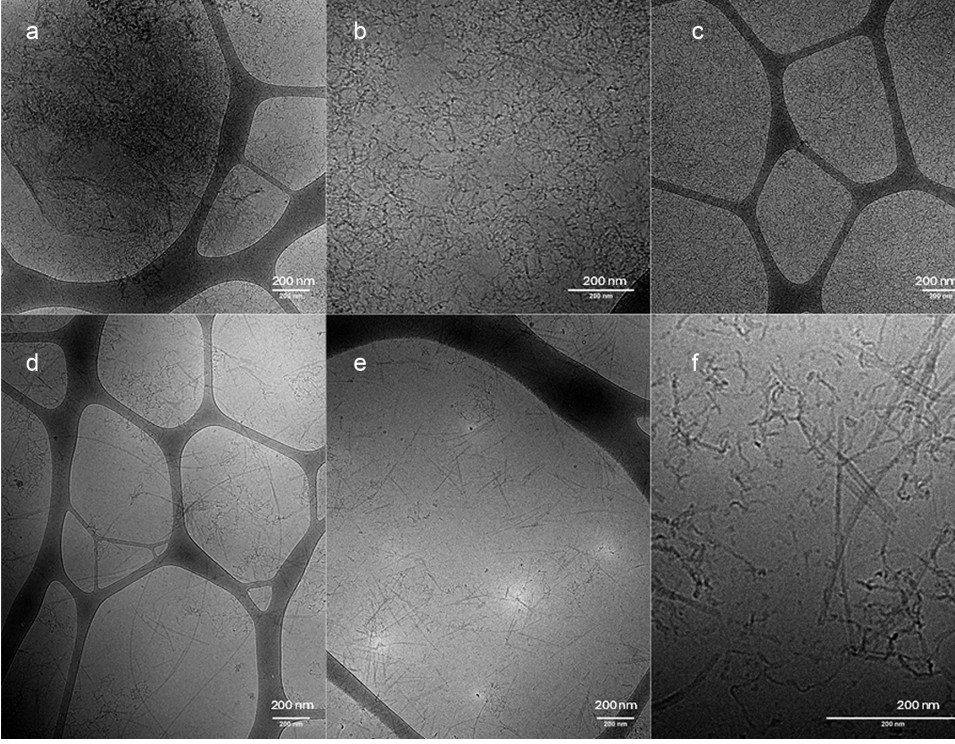

**Figure 1.** Cryo-TEM images of the JB6 linker in phosphate buffer at 25 µM linker (A - C) or 8 µM linker (D - F) incubated at room temperature for various times. Images were recorded immediately after dialysis (A), after 1 week (B), after 2 weeks (C, D, F), and after 5 weeks of incubation (E). The scale bar is 200 nm in all images. (Images were taken at 20000x (A-E), and 80000x (F) magnification).

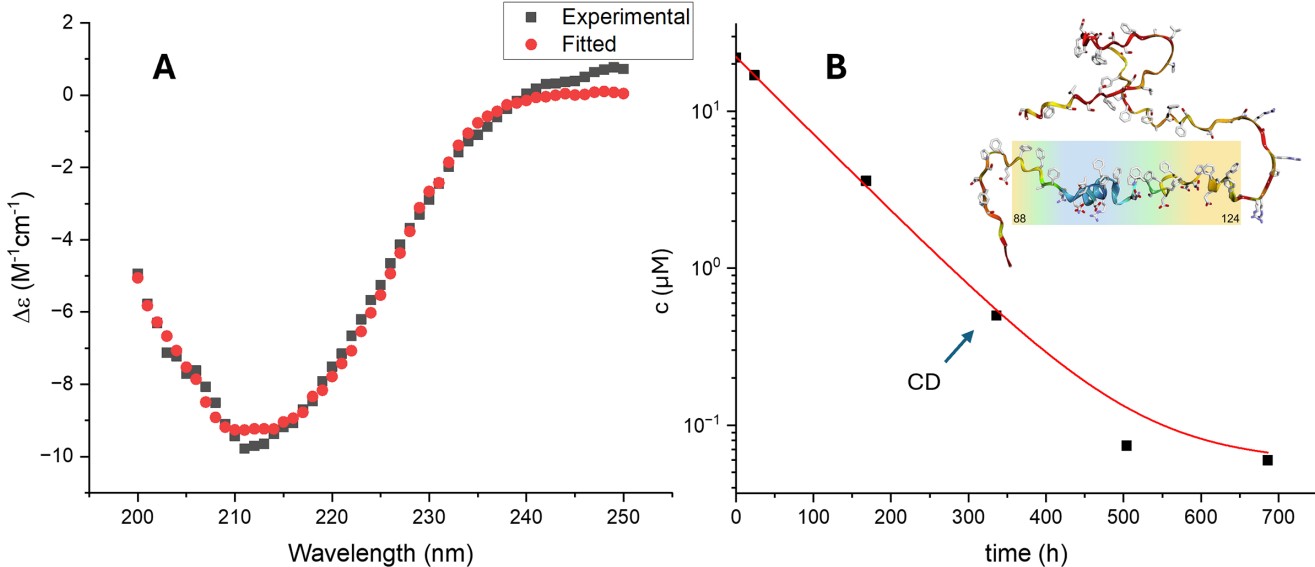

**Figure 2.** A. CD spectrum (black) of the linker sample supernatant after centrifugation after 2-weeks incubation (concentration measured via HPLC). The fit (red) to the data corresponds to 45% α-helix, 20% β-sheet and 35% random coil. The shape of the CD spectrum remains consistent between different samples (Supplementary Info fig. S4). B. The estimated monomer concentration over time from HPLC measurements and a structure prediction from AlphaFold2 (Jumper et al. 2021). The colour coding indicates the structures predicted with yellow for loop and blue for helix. Another measurement was done with a different batch of the linker (Supplementary Info fig. S5). The arrow indicates the time point the CD spectrum was measured.

dissociation from the aggregates is slow compared to the two-week observation time.

Linker aggregates at 2 weeks of incubation at room temperature were studied via FTIR (Supplementary Figure S3). However, the FTIR data are not very clear, possibly due to the different morphologies of aggregates present in the sample.

The CD spectrum of the monomeric linker was recorded to study its secondary structure. This was achieved by measuring the supernatant after centrifugation of the sample as described above.

The spectrum appears to only change in amplitude over time, not in shape, suggesting that the secondary structure of the monomeric linker does not change over time, although aggregates are continuously formed (Figure 2).

By monitoring thioflavin T (ThT) fluorescence over time using a plate reader, Aβ42 fibril formation was evaluated in the presence of different JB6 linker monomer concentrations at 37 °C (Figure 3). In this experiment, a solution of 800 nM monomeric linker diluted from the 25 μM stock in 2 M urea was mixed with

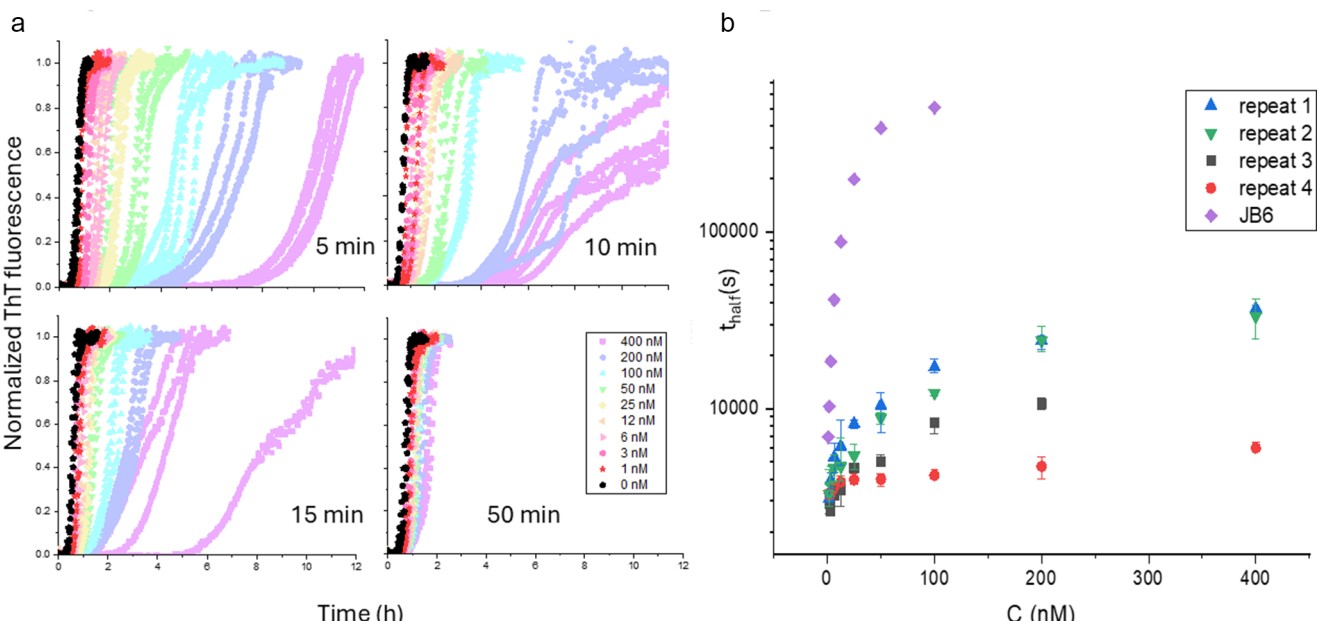

**Figure 3.** A. Linker activity on Aβ42 fibril formation evaluated using thioflavin T fluorescence measurements. The concentration of Aβ42 is 4 μM and the linker concentration is 0 (black) or in the range of 1.5-400 nM (colours, see inset). Four separate experiments with varying preparation time in the range of 5-50 minutes from linker dilution to Aβ42 addition are shown. B. The aggregation half times from the four linker activity experiments in panel A and an analogous experiment for intact JB6 (Linse 2022).

8 µM Aβ42 to yield a solution of 4 µM Aβ42 and 400 nM linker, which was further diluted in a 1:1 volume ratio with 4 µM Aβ42, resulting in a dilution series of the linker at a constant Aβ42 concentration. The most concentrated, 400 nM, linker solution at 1:10 molar ratio of linker to Aβ42 monomer contains 32 mM urea. The effect of urea on fibril formation in this concentration is negligible (Weiffert *et al.*, 2022). A retarding effect can thus be ascribed to the linker and is seen even at 1.5 nM linker. At linker monomer concentrations of 100 nM and 200 nM the fibril formation is significantly retarded, with times of half completion of 2.5–3 hours compared to 40 minutes in the absence of linker. While the effect is not as strong as for the full-length protein, the activity of the linker was found to depend strongly on the time used for the experimental setup. In Figure 3a,b, we compare four experiments with the time lapse from preparing the linker dilution series to the addition of Aβ42 and start of the data collection, ranging from 5 to 50 minutes. Clearly, the activity of the linker is highest at the shortest preparation times. To reach even shorter times, a follow-up experiment was devised, with the linker being diluted to 200 nM at different time points before adding 8 µM Aβ42 to give a final concentration of 100 nM (Supplementary Figures S6–S7). From these results, it appears that at the shortest times after dilution, the linker retards the aggregation greatly. Thus, when the linker is not given time to aggregate, its activity approaches that of the full-length protein, with aggregation half times up to 30 h, which is ca. a factor of 3 shorter than in the presence of 100 nM full-length JB6 (Figure 3*b*). The linker diluted for shorter times before adding Aβ42 also appears to produce aggregates that are significantly less ThT positive (Supplementary Figure S8). A similar experiment was done with 100, 70, and 40 µM α-synuclein with and without the presence of the JB6 linker at 4 µM, showing significant delay in aggregation times (Supplementary Figure S9).

The activity of the fully aggregated linker was investigated by the same method (Supplementary Figure S10). The 25 µM linker sample was incubated at room temperature in phosphate buffer for 4 weeks, resulting in a monomer concentration of around 60 nM. This sample was then diluted and studied at 12–200 nM as above, resulting in linker monomer concentrations of 30–500 pM. No retarding effect was observed. Since the linker appears to form fibrillar aggregates, the ThT fluorescence enhancement of its aggregates was assessed. The ThT positive aggregates seemed to form only in either concentrations orders of magnitude higher than those used in activity experiments or at time scales irrelevant to activity measurements (Supplementary Figure S9). Furthermore, a ThT titration experiment shows that after linker fibrils form, their concentration does not change significantly over the experiment's time period of 4 weeks (Supplementary Figure S12).

## Discussion

The results of this study clearly show that the 110-residue-long linker of JB6 is a potent inhibitor of amyloid formation by Aβ42 from Alzheimer's disease. Still, the full-length protein is significantly more potent, an aspect that we seek to understand through detailed investigations of linker solubility and structure and how these properties differ from the intact JB6.

The linker appears to be partially folded with more than half of its residues in α-helix or β-sheet conformation (Figure 2). The cryo-TEM experiments further suggest that the linker goes through a few

stages of aggregation. Huge clusters form at first; however, these aggregates change with time into either squiggles or a fibril-like state. Since the squiggles dominate in number over the fibril-like aggregates, they appear to be the kinetically favoured form. However, the amount of fibril-like aggregates does not seem to increase relative to the squiggles over time, and both types of aggregation are effectively irreversible within the time frame of the experiments. The appearance of fibril-like structures from the JB6 linker suggests that this part of the protein is amyloid-like. Three prediction algorithms, Tango (Fernandez-Escamilla *et al.*, 2004; Linding *et al.*, 2004; Rousseau *et al.*, 2006), Waltz (Maurer-Stroh *et al.*, 2010) (both predicting the likelihood of β-sheet aggregation in parts of the sequence), and CamSol (predicting solubility in the sequence in arbitrary units) (Sormanni *et al.*, 2015, 2017) were used to predict the solubility and likelihood of aggregation. Two of the algorithms found a common region in the G/F domain of the linker, and all three found a common sequence in the S/T region (Figure 4). Both these parts contain three highly hydrophobic phenylalanine residues, and they may be key parts for both linker aggregation and attractive interactions with amyloid proteins, which is a case for further investigation.

The low complexity linker of DNAJB6 is rather unique among chaperone proteins, compared to disordered regions of other proteins with similar function, such as αB crystallin; the linker is extremely long. DNAJB6 shares its long linker, however, with DNAJB8. These proteins are extremely similar, sharing very similarly structured domains and linkers of almost the same length (107 and 110 amino acids for JB8 and JB6, respectively (Ryder *et al.*, 2021)). The proteins share a similar function in preventing amyloid aggregation, and both also contain G/F and S/T domains. There are key differences, however, shown in Supplementary Figure S13. The length of G/F rich and S/T regions differs; both comprise half of the linker in JB6, and only 37 amino acids make up the S/T domain for JB8. The amino acid composition also differs. JB6 contains more glycine and phenylalanine residues in general, even though the G/F region is shorter, while JB8's contains a wider range of amino acid residues that are not found at all in JB6's linker. JB6's linker contains more charged residues in the G/F region as well. This leads to a difference in the length and hydrophobicity of the two domains between the linkers. Thus, the linker may act as a surfactant with a hydrophobic S/T part and hydrophilic G/F part, leading the intact protein to form aggregates of different geometries (lamellar in the case of JB8 and micelle-like in the case of JB6). Furthermore, using the same prediction algorithms as with the JB6 linker for JB8, a similar aggregation-prone/low solubility region is predicted by Waltz (residues 147–152) and CamSol (residues 151, 153–154).

The ThT fluorescence data clearly show that the linker has a retarding effect on amyloid fibril formation of Aβ42 *in vitro*, at sub-stoichiometric linker:Aβ42 ratios, although the effect is considerably weaker than for the full-length JB6. In addition, the linker has a clear effect on α-synuclein aggregation at sub-stoichiometric linker:α-synuclein ratios (Supplementary Figure S9). Considering that the fibril-like linker may be the active part in attractive chaperone-amyloid interactions, the lower potency of the linker fragment may be explained by the higher propensity of the linker to aggregate with itself. While the full-length protein forms small aggregates that easily dissociate into monomers, the linker forms significantly larger aggregates that dissociate much more slowly and do not dissociate at all over the time scale of the fibril-formation experiments. This results in two competing equilibria: linker self-association and linker-Aβ42 association. The lack of effects of linker aggregates on Aβ42 fibril formation indicates that the

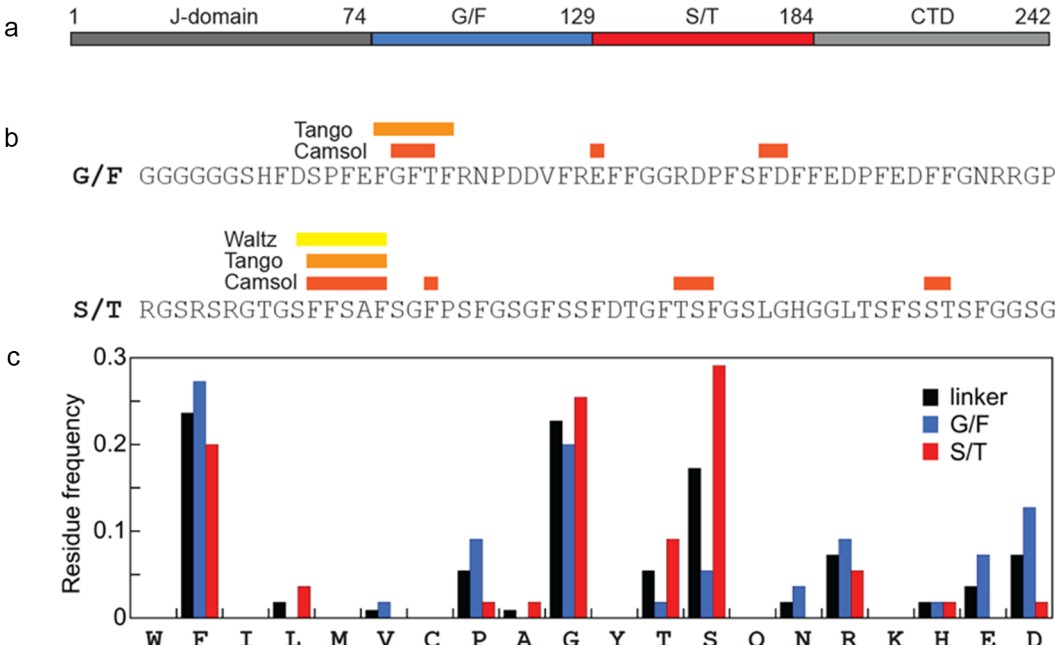

**Figure 4.** A. The different regions of DNAJB6. B. Parts of the linker protein predicted to be aggregation prone by different algorithms. C. The frequency of amino acids in the linker and the G/F and S/T regions in DNAJB6.

dissociated form of the linker is indeed the active part in amyloid inhibition, in analogy with subdomains being the active part of the full-length chaperone (Carlsson *et al.*, 2024).

An outline of a possible molecular mechanism for amyloid suppression by intact JB6 and the linker is presented in Figure 5.

The linker region, which constitutes around half of the full-length protein, is relatively readily available to interact with amyloids in JB6 due to its higher solubility and rapid dissociation at 37 °C (Carlsson *et al.*, 2024). In contrast, the high affinity of the linker fragment towards itself causes the formation of clusters, which

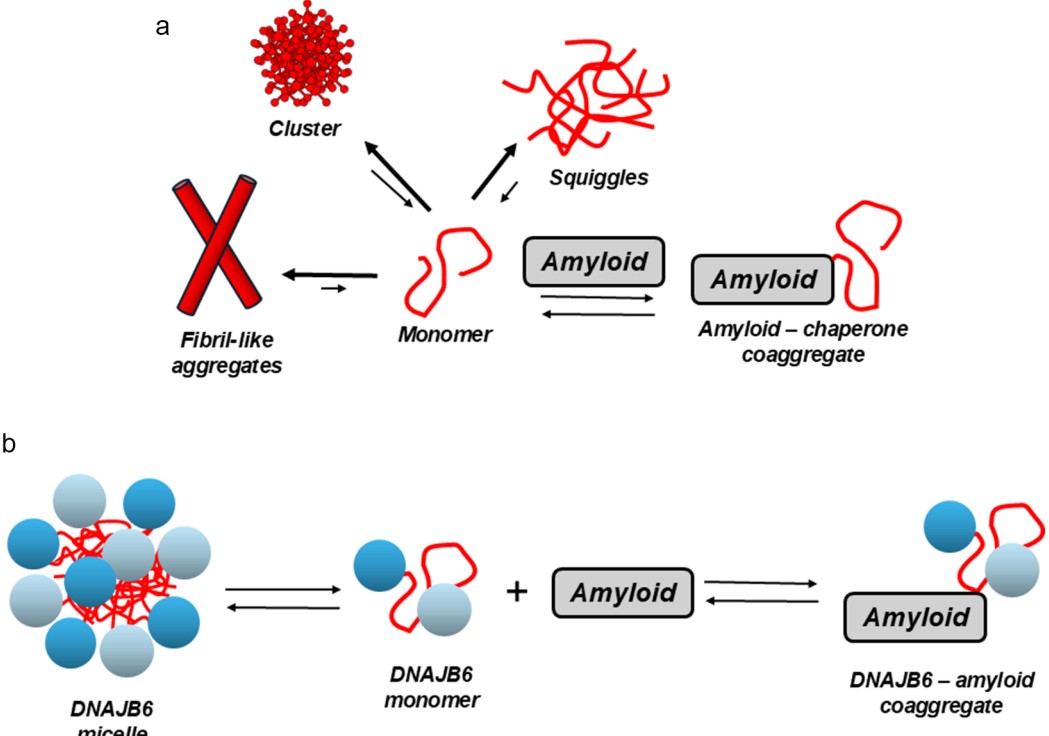

**Figure 5.** A possible mechanism behind the difference in fibril formation retardation efficacy between the linker (A) and intact JB6 (B). While JB6 dissociates readily into monomers at 37°C, which can then interact with amyloid proteins, the linker aggregates dissociate extremely slowly. Multiple competing equilibria involving linker self-aggregates and linker-amyloid protein co-aggregates, result in the requirement for more linker to slow down the aggregation to the same extent as the full-length protein.

dissociate on a relatively short time scale, as well as fibril-like structures and squiggles, both of which dissociate very slowly. In samples of client and monomeric linker (Figure 5a), multiple competing equilibria are created, preventing most of the linker protein from interacting with clients, which results in a weaker fibril retardation effect as opposed to the full-length chaperone. This means that the linker is the key part in amyloid-chaperone interactions and that at least part of the function of the CTD and NTD is to introduce interactions that prevent the formation of squiggles and fibrils, thus keeping the linker more available as an amyloid inhibitor (Figure 5b).

Finally, we ask whether the molecular mechanism revealed here is unique for DNAJB6b or shared with other chaperones. Is the key role of the low complexity linker in amyloid suppression a recurring feature? In the original identification in cells of the anti-amyloid function of JB6 versus poly-glutamine proteins (Hageman et al., 2010), 23 proteins of the Hsp70, Hsp110, and Hsp40 (DNAJ) chaperone families were compared, and such activity was shared with one additional member – DNAJB8. Later, the same function was assigned to DNAJB1 in cells (Ormsby et al., 2013), and activity against tau aggregation has been reported for DNAJB8 (Ryder et al., 2021). DNAJB1 and DNAJB8 have extended low-complexity linkers of 85 and 107 residues, respectively, rich in G, F, S, and T residues but otherwise somewhat higher prevalence of other residue types compared to the JB6 low complexity region.

## Conclusions

The low complexity linker of JB6 is key to its function as an amyloid suppressor. However, the linker is an inferior inhibitor compared to full-length JB6 because of the competing equilibria of linker self-association and the very slow dissociation of linker aggregates. The globular domains of the full protein serve to increase the solubility, the dissociation rate, and the exposure of the linker, making the intact chaperone a more efficient suppressor than the linker alone. The globular domains also serve to increase the aggregate dissociation rate, making the linker in the intact chaperone more readily available as an inhibitor compared to the linker alone.

**Open peer review.** To view the open peer review materials for this article, please visit http://doi.org/10.1017/qrd.2025.10016.

**Supplementary material.** The supplementary material for this article can be found at http://doi.org/10.1017/qrd.2025.10016.

**Data availability statement.** All data presented in this manuscript will be made available upon reasonable request and are uploaded at https://github.com/saralinse/Published_Data/tree/QRB_2025_JB6_linker.

**Acknowledgements.** We would like to thank Crispin Hetherington (nCHERM, Lund University) for the help with grid preparation and cryo-TEM imaging, and Max Lindberg (Biochemistry and Structural Biology, Lund University) for HPLC measurements.

**Author contribution.** T.M., U.O., and S.L. conceived and designed the study. T.M. conducted data gathering. S.L. conducted protein purification. T.M. performed data analyses. T.M. wrote the article with input from the other authors.

**Financial support.** This study was supported by grants from the Knut and Alice Wallenberg Foundation (KAW 2022–0059 to SL and UO), the European Research Council RC (ERC AdG 101097824 to SL) and the Swedish Research Council (VR 2015–00143 to SL and VR 2020–04633 to UO).

**Competing interests.** The authors declare none.

**Inclusion and ethics statement.** This work includes no animals, humans, or samples thereof.

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
