## [Reviewer Report]

The manuscript provides new insights into the mechanism by which the DNAJB6b chaperone inhibits disease-related self-assembly of amyloid peptides in an ATP-independent manner. The authors find that the linker region of DNAJB6 between the NTD and CTD domains (comprising the so-called G/F and S/T regions) forms higher-molecular-weight species and even fibrils. The linker preserves the inhibitory activity of DNAJB6b against Aβ42 aggregation, but not as effectively as the full-length protein. This finding suggests that there is interplay between the domains and the linker region. The authors propose a model in which the linker is key to amyloid-chaperone activity, while the J-domain and CTD stabilize the linker, preventing its aggregation so that it can perform its amyloid-inhibitory function. The publication is methodologically sound, and the experimental design is concise and to the point. Readers will appreciate the thorough explanation of the methodological procedures. We fully support the publication, but we do have a few questions and suggestions that, when addressed, will improve the paper’s clarity and readability.

Major:

I understand that that HPLC allows for monitoring the linker monomeric species and inferring aggregation of the linker, but is the linker ThT positive? I would assume that the authors did a control ThT experiment with just DNAJB66 linker at concentration used in their AB aggregation-prevention-ThT assays. I could not find it in the manuscript, though. Please make sure that it is included in the manuscript. Please also include a sentence in a text about this to ensure that there is no contribution to the fluorescence from the presence of the linker to aggregation of the AB.

Minor:

1. In the results section the authors state “Frozen aliquots of 25 μM linker peptide in 4 M urea were thawed and dialyzed to a 20 mM sodium phosphate buffer, 0.2 mM EDTA, pH 8.0 over 24 h “. Please mention the volume of dialysis or the final concentration of urea.

2. In the results section: After “In the fresh samples, the linker appears to mostly form dense clusters of material ranging from 200 to 1000 nm in diameter”, please add (Fig. 1A).

3. In the results section the authors state “The CTDs are more varied and inferred in client cryo-TEM samples, were centrifuged, to sediment the aggregated material, while the supernatant was collected “. Please indicate at what speed the samples were sedimented.

4. “To study the reversibility of aggregation, an aggregated 25 μM linker sample was diluted 100 times and incubated for two weeks at both 25 and 37 °C. No linker monomer was detected by absorbance in HPLC as above in either sample, measuring several times over the period, implying that the monomer dissociation from the aggregates is slow compared to the two-week observation time.”. Please indicate where these data are presented.

5. In Fig 2. you mark some residues as sticks and color them in yellow/red and introduce a spectrum of colors over the 88-124 sequence. Please explain in the figure legend what the color-coding means.

6. Please add residue numbering when you mention your constructs, for example DNAJB6b (111-222)

7. There is a different citation format in the following statement: The CTDs are more varied and inferred in client interactions14. Please provide the citation

8. In the discussion you say: “There are key differences however, shown in figures 10 and 11” please indicate that there are supplement figures.

---

## [Reviewer Report]

The manuscript by Merkelis et al investigates the linker region of the molecular chaperone DNAJB6 that is a known aggregation inhibitor. Various functions have been assigned to the linker region but relatively little is known about its role experimentally. In this regard the research question of the manuscript is important, but there are various issues that do not allow publication in its current form. The authors focus on the isolated linker peptide and show that it can form fibrillar aggregates. This finding is intriguing, however the full-length protein forms globular aggregates making the relevance of the linker fibrils questionable. In general, it is becoming evident that the linker somehow cooperates with the C-terminal domain to inhibit substrate aggregation and therefore the role of the CTD cannot be ignored. The addition of the linker peptide at different timepoints as shown in Figure 3 is interesting but could be explored further in order to draw more solid conclusions. In fact, this is a major criticism of the manuscript, a lot of the conclusions (including figure 4) seem to be based on data that are too preliminary. Moreover, significant improvements can be made in terms of data presentation and scientific language. Overall, the study deals with a highly interesting question and presents some notable preliminary results that could be the basis for a high quality publication in the future.

---

## [Editor Report]

One reviewer has very serious concerns. If they can be met I am positive to consider publication as I think the Reviewer has not fully taken into account the special character of QRD appreciating originality of new ideas or hypotheses. The other reviewer (signing ‘accept’) has also major comments which I think could be dealt with though.

---

## [Reviewer Report]

Unfortunately, my concerns about the preliminary nature of the results presented have not been addressed in the revised manuscript.

---

## [Reviewer Report]

The authors have addressed my concerns/comments in full, the manuscript is ready for publication in its current state.

---

## [Reviewer Report]

This manuscript describes an investigation into the properties of a 110-residue linker region from the chaperone DNAJB6b, which connects the well-studied J-domain and C terminal domains. The authors have identified that the isolated linker region displays some anti-amyloid activity against Abeta42 in vitro, delaying the onset of Abeta42 assembly. The linker itself self-assembles into aggregates with different morphologies and properties, over time. The anti-amyloid activity of the linker corresponds with the availability or accessibility of monomeric linker. This linker region in DNAJB6 displays some length and compositional similarities with the linker region from other chaperones, notably DNAJB8. The authors propose a mechanism whereby the linker region is required for interaction with amyloidogenic clients, with the globular domains controlling self-assembly and accessibility of the linker region. The results indicate that the linker region may be important in anti-amyloid chaperone activity but the presentation of the work should be improved to increase the value of the work to the research community.

Points to be addressed:

The Abstract contains general statements and multiple references to reviews. This is not standard practice.

What is meant by “JB6 combines a high specificity in action with a high client Promiscuity’? How is high specificity reconciled with client promiscuity?

The manuscript would benefit from careful editing and more attention to the scientific style of writing throughout. Insertion of citations is inconsistent, with some gaps between text and in other places no gaps, and some missing periods, multiple use of “it’s”, loss of correct units (e.g. 4 mL droplet or 4 microlitre droplet used for cryoEM grid preparation?), repeated words (formed), misspellings (e.g. beacue, soubility, aggreagte dissociation) and author comments remaining in SI.

What was the source of the Abeta42, recombinant or synthetic? The source of all reagents should be clearly described.

Why were the experiments performed at room temperature when the activity of the full-length DNAJB6 appears to have been characterised at 37 degree C? There should be further discussion of the effect of temperature on this activity, and the formation of the different linker morphologies.

Is the suggestion that the squiggles and/or fibrils are single sheet structures?

It should be made clear in the main text that the CD signal was collected from the supernatant after centrifugation. What was the structure of the precipitated material? Was this tested for ThT binding, or subjected to FT-IR or X-ray fibre diffraction?

Were the squiggles or fibrils subjected to FT-IR or fibre diffraction analysis? Further structural characterisation of the secondary structure of these alternate forms should be included.

The x-axis should be labelled consistently to facilitate correspondence between experimental data. For example, in Figure 2, convert the x-axis for panel B to weeks, or indicate the time point at which CD samples were analysed.

The results from full-length JB6 are presented for comparison in Figure 3B but appear to have been previously published. Were the data presented here collected at the same time as the linker data? Were they all collected at the same temperature?

How was the higher solubility and rapid dissociation of the linker (referred to in the Discussion) at 37 degree C demonstrated? Is this speculation or have these data been presented?

The activity of the linker towards Ab42 in vitro has been demonstrated. The authors should include experiments with other amyloidogenic proteins to support the generalisation to other amyloids.

Have the authors introduced mutations to regions of the linker and compared activity?

The authors could replace the NTD and CTD with alternative domains to support the suggestion that the function of the CTD and NTD is to “prevent the formation of squiggles and fibrils”.

Consider replacing the terminology “squiggles” with curvilinear structures”.

Consider including Waltz, Tango and Camsol analysis of other chaperone sequences.

The preparation time should be indicated on the panels in Fig. 3A.

The ThT fluorescence data for the linker from 5 weeks should be included.

The data in Fig S10 require further explanation. The legend is not clearly explained.

---

## [Editor Report]

As we received two conflicting reports on this revision, we were obliged to contact a third reviewer to assist us, they have highlighted changes that still need addressing by the auhthors.

---

## [Reviewer Report]

The authors have addressed the major concerns appropriately and some questions will be answered through future experimental work.

Please correct this sentence, appears to be missing a word?

...with the aggregates appearing to take on a tangled coil-like (Fig. 1B).

Add “compared to” into the following sentence: The low complexity linker of DNAJB6 is rather unique among chaperone proteins, compared disordered regions to other proteins of similar function such as αB crystallin...